# Splenic Metastatic Choriocarcinoma with Nontraumatic Splenic Rupture: A Case Report and Literature Review

**DOI:** 10.3390/jcm12010157

**Published:** 2022-12-25

**Authors:** Yifan Chu, Fulan Xu, Zhengguang Ren, Xinyao Hu, Luyao Wang, Jing Yue

**Affiliations:** 1Department of Obstetrics and Gynecology, Tongji Hospital, Tongji Medical College, Huazhong University of Science and Technology, Wuhan 430074, China; 2Department of Obstetrics and Gynecology, Xiaogan Hospital, Wuhan University of Science and Technology, Xiaogan 432000, China; 3Department of Clinical Medicine, Tongji Medical College, Huazhong University of Science and Technology, Wuhan 430074, China

**Keywords:** choriocarcinoma, spleen metastasis, atraumatic splenic rupture, splenic ectopic pregnancy

## Abstract

Choriocarcinoma is a highly malignant trophoblastic tumor that occurs mostly in women of childbearing age. The main mode of metastasis is hematogenous metastasis. The most common sites of metastasis are the lung, vagina and brain, while splenic metastasis is rare. Because of its rapid development, extensive metastasis can occur in a short period, and some patients only show metastatic symptoms, which are often missed or misdiagnosed as ectopic pregnancy or other diseases. We describe a rare case of splenic metastatic choriocarcinoma with acute abdominal pain caused by nontraumatic splenic rupture. In addition, we review the previous literature on splenic metastasis of choriocarcinoma and summarize the clinical manifestations, management measures and prognoses. Our case and literature review indicate that splenic metastatic choriocarcinoma is rare and difficult to distinguish from splenic ectopic pregnancy and other diseases. Clinicians should strengthen their understanding of this disease and avoid misdiagnosis.

## 1. Introduction

Choriocarcinoma is a highly malignant trophoblastic tumor that occurs mostly in women of childbearing age and can be secondary to hydatidiform mole pregnancy or nonhydatidiform mole pregnancy. Its main characteristic is that trophoblasts lose their original villi or hydatidiform mole structure and infiltrate into the myometrium, causing serious local damage. Choriocarcinoma can metastasize to all parts of the body, mainly in the lung, vagina and brain, but rarely in the spleen [1,2,3]. The primary focus of some patients disappeared and they only showed metastatic symptoms, which were often missed or misdiagnosed as ectopic pregnancy and other malignant neoplasms. Splenic rupture is common in trauma, hematological diseases and tumors, whereas nontraumatic splenic rupture and hemorrhage due to splenic metastatic choriocarcinoma are extremely rare, and few surgeons consider the possibility of splenic metastasis of choriocarcinoma when splenic rupture occurs. There are few previous reports about splenic metastasis with splenic rupture in choriocarcinoma. This paper reports a case of splenic metastatic choriocarcinoma with nontraumatic splenic rupture misdiagnosed as splenic ectopic pregnancy and summarizes the experience through a literature review to guide the correct diagnosis and improve therapeutic effects.

## 2. Case Description

A 30-year-old Chinese woman was admitted to a local hospital because of pain in the left upper abdomen for 10 h. A chest and upper abdomen CT showed an apical space-occupying lesion of the upper lobe of the right lung (approximately 4.6 × 4.0 cm, lobulation, burr, with a cavity, considering the possibility of tumor), and multiple splenic nodules (multiple metastatic tumors of the spleen with rupture and subcapsular hematoma were considered). For further diagnosis and treatment, the patient was transferred to the Department of General Surgery, Xiaogan Hospital, affiliated to Wuhan University of Science and Technology. Her past medical history showed that she suffered from hepatitis B. Her menstrual history suggested regular menstruation, but was suspended for 48 days, and her childbearing history included three pregnancies, two deliveries and one miscarriage; the interval from the index pregnancy was 2 years.

After hospitalization, hemostasis, fluid infusion and other symptomatic support treatments were given. Laboratory examination showed a red blood cell count of 3.20 × 10^12^/L, hemoglobin level of 100 g/L, hematocrit of 29.40%, neutrophil ratio of 81.20% and lymphocyte ratio of 15.30%. The tumor markers showed that her serum human chorionic gonadotropin (β-HCG) level was >5000 mIU/mL and her Cyfra21-1 level was 9.78 ng/mL. The T-SPOT.TB assay was negative. An enhanced chest and abdomen CT showed space-occupying lesions in the upper lobe of the right lung (Figure 1), bilateral pulmonary nodules, a small amount of pleural effusion on both sides, multiple splenic space-occupying lesions with subcapsular hematoma (splenic nodules with subcapsular hematoma; the neoplasm was considered) and pelvic and peritoneal effusion. The electrochemiluminescence method detected β-HCG and progesterone quantitatively, while β-HCG levels were 14,328.0 mIU/mL and progesterone levels were 2.48 ng/mL. A transvaginal ultrasound (TVS) showed that endometrial thickness was 1.05 cm, pelvic effusion was 5.45 cm and the uterus, bilateral fallopian tubes and ovaries were normal. There was no obvious abnormality after a bimanual examination by the gynecologist. Because of the increase in β-HCG, the suspicion of ectopic pregnancy was considered. The patient felt that the left upper abdominal pain had been relieved, so we continued conservative treatment. On the eighth day of hospitalization, β-HCG levels were 12,598.0 mIU/mL and progesterone levels were 2.64 ng/mL. A TVS showed that endometrial thickness was 0.90 cm, pelvic effusion was 3.01 cm and the uterus, bilateral fallopian tubes and ovaries were still normal. The patient complained of vaginal spotting, so she was transferred to the department of gynecology.

On the ninth day after hospitalization, dilation and curettage (D&C) were performed. Approximately 10 g of tissue was scraped out of the uterus, without obvious villi, pregnancy tissue or blister-like structures. On the first day after D&C, β-HCG levels were 11,835.0 mIU/mL and progesterone levels were 3.07 ng/mL. The pathological results reported the endometrium to be in the secretory phase. On the third day after D&C, the patient said that the pain in the left upper abdomen was worsening. An ultrasound of the liver, gallbladder, pancreas and spleen showed that a 7.57 × 7.26 cm high echo and irregularly shaped mass had been found on the spleen (Figure 2). The β-HCG level was 16,998.0 mIU/mL and the progesterone level was 3.10 ng/mL. The TVS was still not significantly abnormal. Based on the above medical history and examination results, the suspicion of a splenic ectopic pregnancy with splenic rupture was highly considered.

Emergency laparoscopic exploration was performed after preoperative preparation. During the operation, a large amount of blood accumulated in the abdominal and pelvic cavities, especially in the splenic fossa, and a large number of blood clots could be seen in the splenic fossa. On the surface of the spleen, we found gestational sac-like tissue, which fell off to the omentum majus in the process of exploration, and the tissue was broken in the process of clamping, so no complete specimen was obtained. No obvious focus or bleeding was found in the exploration of the uterus, bilateral fallopian tubes and ovaries, and then a laparotomy was performed. After entering the abdomen, approximately 1000 mL of bloody fluid in the abdominal and pelvic cavities was absorbed. A 4 × 4 cm fissure was seen on the diaphragmatic surface of the spleen, and villous-like tissue was attached around it. Then, the spleen and part of the omentum majus were removed completely (Figure 3A,B). The total blood loss was approximately 1500 mL, and the patient was transfused with 1.5 U of AB-type red blood cells during the operation.

On the third day after the operation, the β-hCG level was 17,694.0 mIU/mL, which was higher than before. Pathological examination suggested metastatic choriocarcinoma of the spleen, and no tumor infiltration in the omentum majus. Immunohistochemistry of the spleen suggested HCG-B (+), Ki-67LI (approximately 95%), PCK (+), PLAP (−), SALL4 (−), CK7 (+), EMA (−), p63 (focus +), P57 (−), CK8/18 (+) and Glypican-3 (+) (Figure 4). Then, the pathological sections were sent to the Department of Pathology of Tongji Hospital, Tongji Medical College, Huazhong University of Science and Technology, for consultation. The results showed that the tumor invaded the spleen. It was composed of abnormal cytotrophoblasts and syncytiotrophoblasts with massive hemorrhage and no chorionic structure, so the diagnosis was splenic metastatic choriocarcinoma, and no tumor cells were found in the omentum majus.

For further treatment, she was transferred to the Department of Obstetrics and Gynecology of Tongji Hospital, Tongji Medical College, Huazhong University of Science and Technology. A chest and abdomen CT showed a mass in the upper lobe of the right lung, and multiple low-density nodules in the liver. According to the 2000 International Federation of Gynecology and Obstetrics (FIGO) Staging and World Health Organization (WHO) scoring system for gestational trophoblast neoplasm (GTN) (Table 1 and Table 2) [4], the final diagnosis of this patient was choriocarcinoma (stage IV: 11 scores). Then, the patient received chemotherapy, which included regimens of etoposide, methotrexate, actinomycin D/cyclophosphamide and vincristine (EMA-CO) for one course, and etoposide, cisplatin/etoposide, methotrexate and actinomycin D (EP-EMA) for three courses. After four courses of chemotherapy, the β-hCG level still did not decrease to normal. We considered the existence of chemotherapy-resistant lesions, so a thoracoscopic right upper lung mass wedge resection and lymph node biopsy were performed. A pathological examination showed pulmonary metastasis of choriocarcinoma. After surgery, the β-hCG level decreased to normal, and four courses of chemotherapy were supplemented with the EP-EMA regimen. There was no sign of recurrence and the patient was still in follow-up.

## 3. Discussion

Choriocarcinoma is a highly malignant trophoblastic tumor that occurs mostly in women of childbearing age and can be secondary to hydatidiform mole pregnancy or nonhydatidiform mole pregnancy. The incidence of choriocarcinoma is 5 to 202 per 100,000 pregnancies in China [5], 1 per 40,000 pregnancies in North America and Europe and 9.2 and 3.3 per 40,000 pregnancies in Southeast Asia and Japan [6], respectively. It is composed of neoplastic intermediate trophoblasts, cytotrophoblasts and chorionic syncytiotrophoblasts, which are characterized by trophoblasts losing their original villi or hydatidiform mole structure and infiltrating into the myometrium of the uterus, causing serious local damage. Choriocarcinoma is prone to early systemic metastasis [2,3]. The tumor can metastasize to all parts of the body, and the most common metastatic sites are the lung, vagina, liver, digestive tract and brain. The clinical manifestations of choriocarcinoma associated with nonmolar gestation are often atypical and can have both primary and metastatic symptoms, or only metastatic symptoms. Aside from postpartum abnormal vaginal bleeding, many patients with choriocarcinoma often present with metastatic symptoms at the first visit, among which cardiopulmonary, digestive and central nervous system symptoms are the most common, including bleeding from metastatic sites such as the liver, spleen, intestines, lung or brain, pulmonary symptoms and neurological signs from spine or brain metastasis [6,7]. The spleen is an infrequent site of secondary visceral metastases and an uncommon site of metastatic choriocarcinoma. Splenic rupture is usually seen in trauma, but atraumatic splenic rupture due to metastatic choriocarcinoma is extremely rare [2,8]. A previous study found that choriocarcinoma accounted for only 2% of splenic metastatic tumors [9]. In recent decades, several cases of splenic metastatic choriocarcinoma have been reported (Table 3). Among them, 12 patients received laparotomic or laparoscopic splenectomy [5,10,11,12,13,14,15,16,17,18,19,20] and 1 patient received splenic artery embolization [21] due to splenic rupture or bleeding, which is consistent with the growth characteristics of trophoblast cells. In addition to splenic metastasis, 11 patients had other organ metastases, such as liver, lung or brain metastasis [5,10,11,12,13,14,17,18,19,20,21]. Six female patients and a male patient were suspected of ectopic pregnancy or lymphoma before pathological examination [5,11,12,13,15,16,17], which also reflects the concealment and high misdiagnosis rate of this disease. Four patients died during follow-up [13,14,16,17], which was related to late diagnosis, multiple metastases and tumor progression.

This patient presented with pain in the left upper abdomen as the first symptom. A chest and abdominal CT showed a pulmonary apical mass, multiple nodules of the spleen and subcapsular hematoma of the spleen. It was our accidental finding of an abnormal elevation in β-hCG levels that led to the consideration of pregnancy-related diseases. The special feature of this patient is that there are no typical reproductive system manifestations of trophoblastic tumors, such as abnormally enlarged uterine volume, vaginal purplish–blue nodules or abnormal intrauterine scrapes, and no obvious abnormalities were found via a TVS. The mass at the tip of the lung is difficult to distinguish from lung tumors and pulmonary tuberculosis, which undoubtedly increases the difficulty of disease diagnosis. Choriocarcinoma has no chorionic villi in histology [6]; however, in the process of laparoscopy and laparotomy, it seemed that gestational sac and villous-like tissue could be seen on the surface of the spleen, which made us empirically misdiagnose it as a spleen ectopic pregnancy. We did not make a definitive diagnosis until we found further increases in β-hCG levels after surgery and after a pathological examination suggested choriocarcinoma. hCG is an excellent biomarker of disease progression, response and subsequent posttreatment surveillance. A plateaued or rising β-hCG level enables the early detection of the progression of a partial and complete hydatidiform mole to GTN [6]. Shinoda et al. suggested that patients with extremely high levels of β-hCG can be diagnosed with choriocarcinoma even before the pathological results are obtained [22], but the cutoff value of β-hCG in the diagnosis of choriocarcinoma associated with nonmolar gestation has not been definitively determined due to the differences in reagents, detection methods and sample dilution times between every laboratory.

The overall cure rate of GTN is currently more than 90% due to the application of chemotherapy and other comprehensive treatments. The best regimen depends on the stage and classification [23]. In the 2000 FIGO staging and classification and the WHO scoring system, a risk score of 6 and below is classified as low risk and a score above 6 is considered high risk [4]. Those among the high-risk group and the ultra-high-risk subgroup with a score of 13 or greater, as well as patients with liver, brain or extensive metastases, have a higher treatment failure rate, and thus a poorer prognosis [24]. Patients with low-risk GTN should be treated with one of the single-agent methotrexate or actinomycin D protocols. After the hCG level has returned to normal, consolidation with 2−3 more cycles of chemotherapy will decrease the chance of recurrence. The overall complete remission rate is close to 100% [2,23]. Multiple-agent chemotherapy regimens are used to treat high-risk GTN, and the most commonly used is EMA-CO. However, approximately 20% of patients do not attain a complete response with EMA-CO therapy, but most can be salvaged with further therapy [6]. For those patients with liver metastases, with or without brain metastases, or a very high-risk score, EP-EMA and other more intensive chemotherapy regimens such as BEP (bleomycin, etoposide, cisplatin), FAEV (floxuridine, actinomycin-D, etoposide, vincristine) and immunotherapy rather than EMA-CO may yield a better response and outcome. For such high-risk patients, a longer consolidation with four cycles of chemotherapy should be considered.

In addition, the significance of surgery cannot be undervalued. A postoperative pathological examination can clarify the diagnosis and avoid misdiagnosis, especially in metastatic choriocarcinoma. Hysterectomy or uterine artery embolization can be considered in cases of uncontrolled uterine bleeding. Laparotomy or laparoscopy may be needed to stop bleeding in organs such as the liver, gastrointestinal tract, kidneys and spleen. Neurosurgery is needed if there is bleeding into the brain or increased intracranial pressure. The resection of an isolated drug-resistant tumor may also be curative [25,26].

Reviewing the diagnosis and treatment of these patients, the experience is as follows: (1) For women of childbearing age and with a sexual history, when abdominal pain, irregular vaginal bleeding and no specific clinical symptoms occur, hCG detection by blood or urine should be performed to rule out pregnancy-related diseases; (2) For patients with elevated β-hCG combined with the occupation of the liver, spleen, lung, brain and other organs, attention should be given to the possibility of choriocarcinoma metastasis even if there are no obvious abnormal manifestations in the reproductive system; (3) Ectopic pregnancy is most common in the fallopian tube. An obvious increase in β-hCG levels is often seen in intrauterine pregnancy, or special cases such as cornual pregnancy and tubal interstitial pregnancy. The possibility of no positive findings in uterine and bilateral fallopian tubes and ovaries via ultrasound examination is very small. Therefore, gestational trophoblastic disease should be highly considered; (4) β-hCG levels should be decreased after resection of the focus in ectopic pregnancy, but if β-hCG further increases after the operation, the possibility of gestational trophoblastic disease should also be considered, and pathological examination is the gold standard for diagnosis; (5) If β-hCG levels cannot be reduced to the normal level after standard chemotherapy, isolated drug-resistant tumor resection can be chosen according to the patient’s condition.

## 4. Conclusions

Splenic metastatic choriocarcinoma with splenic rupture and bleeding is very rare, the onset is acute and hemorrhagic shock can occur in a short period of time and even be life-threatening. These patients should be identified and treated as soon as possible. Dynamic monitoring of β-hCG levels, ultrasound and CT/MRI are helpful for early diagnosis. Once diagnosed, most patients should undergo an operation to save their lives, followed by standardized chemotherapy. The treatment of patients with drug resistance and relapse should focus on the development of individualized treatment plans according to the degree of disease and previous treatments. In addition to active chemotherapy, surgery can also be used to remove drug-resistant or recurrent foci. After reaching a full course of chemotherapy, patients still need close follow-up and regular reexamination of blood β-hCG levels, ultrasound and CT/MRI to observe the recurrence of choriocarcinoma.

## Figures and Tables

**Figure 1 jcm-12-00157-f001:**
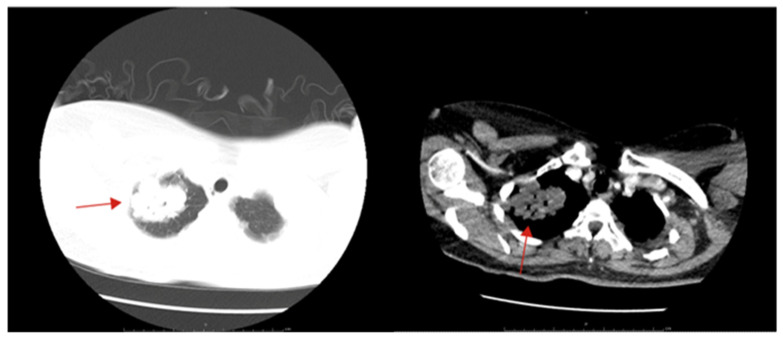
Chest CT showed a mass in the right upper lung (solid red arrow).

**Figure 2 jcm-12-00157-f002:**
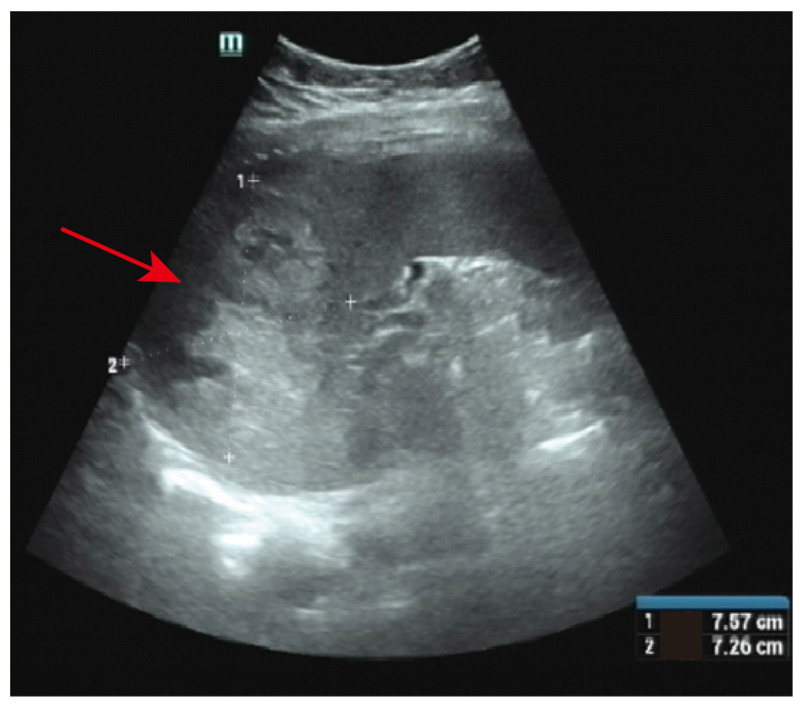
Ultrasound showed abnormal hyperechoic areas in the spleen (solid red arrow).

**Figure 3 jcm-12-00157-f003:**
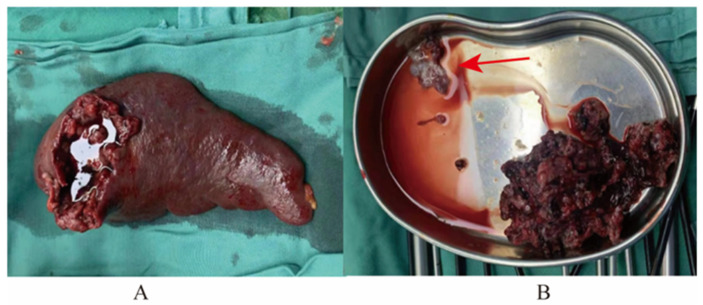
Specimens obtained in the operation. (**A**) Spleen. (**B**) Villous-like tissue (solid red arrow) and other tissues.

**Figure 4 jcm-12-00157-f004:**
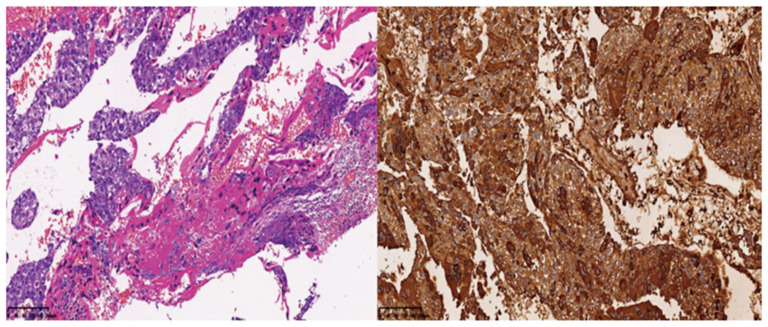
Pathological image of the spleen and immunostaining of hCG.

**Table 1 jcm-12-00157-t001:** FIGO staging and classification for GTN.

FIGO Stage	Description
I	Gestational trophoblastic tumors strictly confined to the uterine corpus.
II	Gestational trophoblastic tumors extending to the adnexa or to the vagina, but limited to the genital structures.
III	Gestational trophoblastic tumors extending to the lungs, with or without genital tract involvement.
IV	All other metastatic sites.

**Table 2 jcm-12-00157-t002:** WHO scoring system based on prognostic factors modified as FIGO score.

Score	0	1	2	4
Age (years)	<40	>40	-	-
Antecedent pregnancy	Mole	Abortion	Term	-
Interval from index pregnancy (months)	<4	4–<7	7–12	>12
Pretreatment hCG (IU/L)	≤10^3^	>10^3^–10^4^	>10^4^–10^5^	>10^5^
Largest tumor size (including uterus, cm)	-	3–<5	≥5	-
Site of metastases	Lung	Spleen, kidney	Gastrointestinal tract	Liver, brain
Number of metastases identified	-	1–4	5–8	>8
Previous failed chemotherapy	-	-	Single drug	Two or more drugs

**Table 3 jcm-12-00157-t003:** Reported Cases of Splenic Metastatic Choriocarcinoma.

	Author	Age/Sex	Symptom	Operation	Metastatic Organs	Suspicion	Prognosis
1	Zhang et al. [5]	39 y/F	AP, VB	S, resection of omental, liver, lung metastases	spleen, liver, omentum majus, lung	SMC/EP	Continuous remission
2	Kristoffersson et al. [10]	31 y/F	SP	S	spleen, ileum	unknown	Recovered
3	Satoko et al. [11]	31 y/F	AP, hypotension	S	spleen, brain, lung	EP	Relieved
4	Challis et al. [12]	30 y/F	AP, SP, VB, nausea, shock	S, D&C	spleen, liver	EP	Relieved
5	Ghinescu et al. [13]	71 y/M	AP, sweat, nausea	S	spleen, liver, lung	lymphoma	Died
6	Luu et al. [14]	23 y/F	AP	S	spleen, brain	unknown	Died
7	Yuruyen et al. [15]	29 y/F	AP, nausea, vomit	S	spleen	NHL	No recurrence
8	Mukuku et al. [16]	38 y/F	AP	S, D&C	spleen	EP	Died
9	Aloysius et al. [17]	41 y/F	AP, vomit, melena	S, D&C	spleen, liver, lung, brain, duodenum, proximal jejunum	EP	Died
10	Hou et al. [18]	32 y/F	AP	S	spleen, liver, kidney	unknown	Recovered
11	Ramarajapalli et al. [19]	25 y/F	AP, backache	OTR, S, A, D&C	ovary, spleen, adrenal gland	unknown	Complete remission
12	Nethra et al. [20]	37 y/F	AP, SP, dizzy	S	spleen, lung, kidney	GTD	No recurrence
13	Galazi et al. [21]	28 y/F	AP, shortness of breath, headache	E	spleen, liver, lung, brain	unknown	No recurrence

Abbreviations: AP: abdominal pain, SP: shoulder pain, VB: vaginal bleeding, S: splenectomy, D&C: dilation and curettage, E: embolization, OTR: ovary tumor resection, A: adrenalectomy, NHL: non-Hodgkin lymphoma, EP: ectopic pregnancy, GTD: gestational trophoblast disease, SMC: splenic metastatic choriocarcinoma.

## Data Availability

Not applicable.

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
