# Peer review of "Splenic Metastatic Choriocarcinoma with Nontraumatic Splenic Rupture: A Case Report and Literature Review"

_jcm, 2022, doi:10.3390/jcm12010157_

Round 1

Reviewer 1 Report

The authors describe a case report and review of the literature on Splenic Metastatic Choriocarcinoma with Non-traumatic Splenic Rupture, which is a rare, but interesting topic, adding knowledge to the rare topic of choriocarcinoma. The paper is well structured and written, but it would profit from a language editing. The conclusions are consistent with the evidence and arguments presented and they address the main question posed. Besides the authors should improve the quality of their figures (CT scans). I therefore recommend minor revisions.

Author Response

Dear Reviewer,

Thank you very much for your time involved in reviewing the manuscript and your very encouraging comments on the merits.

The authors describe a case report and review of the literature on Splenic Metastatic Choriocarcinoma with Non-traumatic Splenic Rupture, which is a rare, but interesting topic, adding knowledge to the rare topic of choriocarcinoma. The paper is well structured and written, but it would profit from a language editing. The conclusions are consistent with the evidence and arguments presented and they address the main question posed. Besides the authors should improve the quality of their figures (CT scans).

We also appreciate your clear and detailed feedback and hope that the explanation has fully addressed all of your concerns. In the remainder of this letter, we discuss each of your comments individually along with our corresponding responses.

Point 1: The paper is well structured and written, but it would profit from a language editing.

Response 1: Thanks for your great suggestion. We have carefully polished the language by the editing service to improve readability of the revised manuscript.

Point 2: Besides the authors should improve the quality of their figures (CT scans)

Response 2: Thank you for the detailed review. We have updated the CT scans figures with higher quality in the revised manuscript.

We would like to thank the referee again for taking the time to review our manuscript.

Reviewer 2 Report

The authors describe an extremely rare case of unusual splenic choriocarcinoma with rupture. And they emphasized that this case and literature review indicates that splenic choriocarcinoma is difficult to distinguish from splenic ectopic pregnancy and other disease.

It might be improved on the following issues.

 In this case report, differential diagnosis is very important. Especially, the most important point of this article is the pathological diagnosis of the splenic lesion.

#1. In the “Title”: I think “Splenic choriocarcinoma with Non-traumatic Splenic Rupture, suspicious of metastatic”. See item #5 and #6 below

#2. P.2, line 51, “Menstrual history”: When was the last menstruation?

#3. P.2, line 52, “childbearing history”: How long was the interval from the index pregnancy?

#4, P.2, line 63, “Immunohistochemistry”: Which organ(tissue) was performed on?

#5, P.3, line 101, “villous tissue”: If there was a coexistent fetus, this is the fetal comorbid choriocarcinoma. It is necessary to prepare and examine enough number of sections for pathologic diagnosis. How many blocks were taken from the splenic lesion?

#6. P.4, line 112, “pathological sections”: The microphotographic evidence of diagnosis is required. The splenic choriocarcinoma which has pleomorphism of trophoblastic tumor cells with hemorrhage and necrosis. And if possible, please show the photograph of immunostaining for hCG.

Author Response

Dear Reviewer,

Thank you very much for your time involved in reviewing the manuscript and your very encouraging comments on the merits.

The authors describe an extremely rare case of unusual splenic choriocarcinoma with rupture. And they emphasized that this case and literature review indicates that splenic choriocarcinoma is difficult to distinguish from splenic ectopic pregnancy and other disease.

We also appreciate your clear and detailed feedback and hope that the explanation has fully addressed all of your concerns. In the remainder of this letter, we discuss each of your comments individually along with our corresponding responses.

Point 1: In the “Title”: I think “Splenic choriocarcinoma with Non-traumatic Splenic Rupture, suspicious of metastatic.

Response 1: Thanks for your great suggestion. We agree with the reviewer that suspicious metastatic can added to the title of the manuscript. However, we think the original title is more appropriate based on the results of pathological examination, immunohistochemistry and no abnormal reproductive system. For this reason, we chose not to make this change.

Point 2: P.2, line 51, “Menstrual history”: When was the last menstruation?

Response 2: Thank you for the detailed review. The patient's last menstruation was on December 10, 2021, and the hospitalization time was on January 27, 2022. Until hospitalization, the patient had been menopause for 48 days. We have supplemented this information in the manuscript (P.2, line 52).

Point 3: P.2, line 52, “childbearing history”: How long was the interval from the index pregnancy?

Response 3: Thank you for the detailed review. The patient's last pregnancy occurred in 2019, and the interval from the index pregnancy was 2 years. We have supplemented this information in the manuscript(P.2, line 54).

Point 4: P.2, line 63, “Immunohistochemistry”: Which organ(tissue) was performed on?

Response 4: Thank you for the detailed review. We quantitatively measured hCG and progesterone in patients' peripheral blood by electrochemiluminescence method, but made a mistake in writing as immunohistochemistry. We have corrected this error(P.2, line 64).

Point 5: P.3, line 101, “villous tissue”: If there was a coexistent fetus, this is the fetal comorbid choriocarcinoma. It is necessary to prepare and examine enough number of sections for pathologic diagnosis. How many blocks were taken from the splenic lesion?

Response 5: Thank you for the detailed review. We were very careful in the pathologic diagnosis process. Five tissues were removed from the spleen and 20 pathological sections were made. No pregnancy tissue such as pregnancy sac or villi was found in each section. The superior hospital got the same conclusion after pathology consultation.

Point 6: P.4, line 112, “pathological sections”: The microphotographic evidence of diagnosis is required. The splenic choriocarcinoma which has pleomorphism of trophoblastic tumor cells with hemorrhage and necrosis. And if possible, please show the photograph of immunostaining for hCG.

Response 6: Thank you for the detailed review. We have added pathological examination and hCG immunohistochemical figures in the manuscript(Figure 4).

We would like to thank the referee again for taking the time to review our manuscript.